# Difficulties in Accessing Cancer Care in a Small Island State: A Community-Based Pilot Study of Cancer Survivors in Saint Lucia

**DOI:** 10.3390/ijerph18094770

**Published:** 2021-04-29

**Authors:** Aviane Auguste, Glenn Jones, Dorothy Phillip, James St. Catherine, Elizabeth Dos Santos, Owen Gabriel, Carlene Radix

**Affiliations:** 1Vaughan Arthur Lewis Institute for Research and Innovation (VALIRI), Sir Arthur Lewis Community College, Morne Fortune, Castries LC06 101, Saint Lucia; stcatherine1950@gmail.com; 2Faculty of Medical Sciences, The University West Indies, Mona, Kingston 7, Jamaica; gwjones6@live.com; 3Faces of Cancer Saint Lucia, Tapion Ridge, Castries LC04 201, Saint Lucia; facesofcancerslu@gmail.com; 4Organisation of Eastern Caribbean States (OECS) Commission—Franck Johnson Avenue, Morne Fortune, Castries LC06 101, Saint Lucia; edsantos@bvihsa.vg (E.D.S.); carlene.radix@oecs.int (C.R.); 5Department of Oncology, Owen King European Union Hospital, Millenium Highway, Castries LC04 201, Saint Lucia; gabrielowen2000@gmail.com; 6Caribbean Association for Oncology and Hematology, Belmont, Port of Spain 150123, Trinidad and Tobago

**Keywords:** cancer, health care delivery, medical tourism, care pathways, community health, health disparities, Caribbean, Saint Lucia, small island developing state, low-and-middle income country

## Abstract

Developing robust systems for cancer care delivery is essential to reduce the high cancer mortality in small island developing states (SIDS). Indigenous data are scarce, but community-based cancer research can inform care in SIDS where formal research capacity is lacking, and we describe the experiences of cancer survivors in Saint Lucia in accessing health services. Purposive and snowball sampling was used to constitute a sample of survivors for interviews. Subjects were interviewed with a questionnaire regarding socio-demographics, clinical characteristics, health services accessed (physicians, tests, treatment), and personal appraisal of experience. We recruited 50 survivors (13 men, 37 women). Only 52% of first presentations were with general practitioners. The mean turnaround for biopsy results in Saint Lucia was three times longer than overseas (*p* = 0.0013). Approximately half of survivors commenced treatment more than one month following diagnosis (median of 32 days, IQR 19–86 days), and 56% of survivors traveled out-of-country for treatment. Most survivors (60%) paid for care with family/friends support, followed by savings and medical insurance (38% each). In conclusion, cancer survivors in Saint Lucia are faced with complex circumstances, including access-to-care and health consequences. This study can guide future research, and possibly guide practice improvements in the near term.

## 1. Introduction

Small island developing states (SIDS) are particularly affected by cancer because of their inherently fragile health care systems secondary to smaller populations, fewer resources, and vulnerability to natural disasters [1,2,3]. Further, SIDS experience a higher mortality-to-incidence ratio compared with more developed islands in their region [1,3]; this alludes to system-contingent cancer mortality [4,5]. SIDS might reduce cancer burden by providing their populations with comprehensive cancer care services. Models of cancer care in SIDS remain heavily orientated towards out-bound medical tourism, which refers to travel overseas for care. In some situations, imported or overseas services are carefully coordinated and subsidized by governments, especially in developing countries in the Asia and Pacific regions [6,7]. However, other SIDS have less developed models that also intensify health disparities [1,3]. Developing robust systems for cancer care delivery around out-bound medical tourism may be a solution to improve access to care, given the inherent fiscal and capacity limitations of SIDS [8,9]. 

Saint Lucia is a SIDS in the Caribbean with approximately 180,000 inhabitants, mostly of Afro-Caribbean descent. Between 2008 and 2012, Saint Lucia recorded 478 cancer deaths among men and 420 among women (age-standardized rate per 100,000, M: 98.8 and F: 79.6) [10], with incidence perhaps double that; so, cancer is a human burden in the country. Although universal health coverage is not yet established, the country possesses a national social security system that contributes partly to health care costs. The country possesses five hospitals and numerous health centers. The best equipped facilities are in the capital (Castries), though at reasonable distance from smaller towns [11]. In terms of cancer care, patients have access to basic services for diagnosis, surgery, and chemotherapy. In certain cases, tertiary cancer care services are accessible only abroad (e.g., radiotherapy) [12], with limited financial support from local health authorities [1]. Regarding prevention and control, there is no comprehensive capacity for cancer control or population-based surveillance; however, there is a national non-communicable disease (NCD) plan, and a hospital-based cancer registry activity [1].

Currently, high-level system indicators have been identified for Saint Lucia [1,12], but data on patient experiences accessing cancer care and related factors and processes are lacking. These knowledge gaps cannot be addressed locally using the same approach as previous studies from high-income countries [13,14,15,16,17,18,19,20,21,22,23,24,25] given the limited research capacity on the island, which is a common issue among SIDS [26]. On the other hand, community-based approaches in research have been successfully implemented in some LMICs to generate responsive and patient-centered insights into cancer prevention [27,28,29]. A study of this type is novel for Saint Lucia. This approach could drive sustainable indigenous research on cancer and other non-communicable diseases in Caribbean SIDS.

Herein, we report findings from a community-based study referred to as DCAP (Description of the Cancer Health Services: Diagnosis and Treatment Pathways). This study aimed to describe, for the first time, experiences in accessing cancer health services by cancer survivors residing in Saint Lucia, and to explore determinants of access to care. Herein, we also report our community research approach, and include lessons learned during the implementation of this study.

## 2. Materials and Methods

### 2.1. Community-Based Collaboration

This work was the product of novel collaboration, in Saint Lucia, between physicians, academic researchers, public health administrators, cancer advocates, and cancer survivors from the community. The main organizations were from Saint Lucia: Faces of Cancer Saint Lucia (FOCS), a non-governmental organization (NGO); the Vaughan Arthur Lewis Institute for Research and Innovation (VALIRI); and the Organization of Eastern Caribbean States (OECS). The last two both have experience in public health research (Appendix A). The OECS provided the financial resources.

### 2.2. Patient Recruitment 

We constituted a retrospective convenience cohort of cancer survivors between May 2019 and August 2020. Eligible patients were greater than 18 y of age, able to communicate in English or Creole (without cognitive impairment), with an invasive cancer diagnosis (any cancer site, histology, and year of diagnosis), and having accessed health services in Saint Lucia due to cancer. Participation included authorization to access a patient’s data from medical records in health care institutions and centers. Sources for subject recruitment were FOCS, Victoria Hospital, the Oncology center, and key informants. Patients at health care establishments were recruited during opportunistic cancer navigation assistance by a FOCS representative. Key informants were recruited using purposive sampling [30]. When possible, we recruited key informants during cancer advocacy activities organized by FOCS. Snowball sampling was used during interviews to identify prospective participants [30,31]. We screened data sources for potentially eligible participants then invited as many patients as possible. Next-of-kin were interviewed where the index patient was deceased, or not physically able to undergo an interview. 

### 2.3. Ethics

The DCAP study was granted ethics approval by the ethics committee from the Medical and Dental Council (Saint Lucia, WI) in April 2019. All participants provided written informed consent prior to the study-required interview. Completed questionnaires contained no nominative information and were stored separately from consent forms. Further, all data were anonymized prior to transfer to investigators, thus having no personal identifiers.

### 2.4. Data Collection and Questionnaire

Eligible participants were interviewed face-to-face by trained field investigators using a single, standardized questionnaire (Appendix A). Participants were asked to have on-hand their test reports and personal clinical documents, to use as memory-aids during interviews. 

The questionnaire was developed to ascertain sociodemographic variables such as education level, private medical insurance, hot water at home, employment/education, and clinical characteristics, such as cancer stage at diagnosis, and comorbidities. We also obtained detailed information from all consultations with health care providers (HCP), from first presentation to initiation of active treatment, including investigations, specialty of the HCP, location of consultation, scheduling and actual date of consultation, symptoms, tests and treatments prescribed, referrals, scheduling of review appointments, and patient/family suggestions for improving the described management. For each diagnostic test, date, lab name and location, turnaround time, and the finding (whether or not a test revealed suspicion of cancer) were obtained. Date of specimen collection was the date ascribed for blood and histology tests. All treatment modalities were recorded, including natural/alternative remedies. For each modality, information from the first and last time it was administered (type, date, specialty of physician, location, and country) were recorded. Given a context of “outbound medical tourism” in Saint Lucia, we designed a section of the questionnaire specific to those who accessed tests and/or treatment overseas. For every country visited for diagnosis and/or treatment, we recorded services accessed, and personal patient/family motives behind these choices. Information on psycho-social support, supportive care services, experience in obtaining funds (for cancer care), palliation, symptom control, post-treatment follow-up, and prevention were collected during the interview. Participants’ personal appraisal of their experiences for major events was ascertained throughout the interview. Any other miscellaneous remarks pertaining to cancer experience were recorded.

### 2.5. Variables and Definitions

Definitions for dates of first symptoms, first presentation, and referrals were per the Aarhus statement [32]. Definitions for time intervals between key events were adopted from the model of pathways to treatment [33]. In cases where cancer was discovered incidentally (e.g., screening), date of first presentation was assigned as the date of the consultation/test/specimen collection allowing for first detection of the cancer. Reason to consult with HCP refers to the event/action that compelled a participant to discuss the body changes/symptoms with a healthcare professional. Date of diagnosis was determined solely through self-report from the question “when were you first diagnosed with cancer”. Initiation of first treatment was defined as the time point at which the first active cancer treatment was administered by a HCP, regardless of modality. Diagnosis abroad was defined as a medical test performed that required physical travel outside of Saint Lucia. Treatment abroad was defined as a therapeutic intervention administered abroad.

We did not measure income directly, but used private health insurance and hot water at home as surrogate indicators [34]; these two variables correlated with educational level (Appendix A). Last, Appendix A provides definitions for all other study variables.

### 2.6. Preparation and Implementation

A steering committee constituting main stakeholders outlined objectives and the design of the study, and oversaw implementation. The protocol and study questionnaire were drafted and then approved by all members. Two test interviews were conducted with volunteer cancer survivors from FOCS. Then, minor ergonomic revisions were made to the questionnaire, submitting the final protocol to the local research ethics committee.

Following ethics approval, student volunteers prepared materials and supplies to facilitate interviews. Potentially eligible participants were pre-screened, to be approached. Trained field investigators provided regular updates on participant inclusions and operational challenges. In addition, they rated the quality of every interview from one (poor) to five (excellent) based on the cooperation of the patient and the quality of responses.

### 2.7. Data Analysis

Data were entered into MS Access. Missing or incomplete responses for questions on dates were dealt with according to available information from interviews; if no additional information was available, imputation was undertaken to ensure coherence (e.g., dates for which the day was missing only, were replaced by the 15th day of the month, and dates with missing day and month were replaced by 1st of January of the noted year). Descriptive summary statistics included value distributions. Median time intervals and interquartile ranges, and mean turnaround times with standard deviations (SD) for test results were determined. Student *t*-tests were conducted for mean time difference between tests done in Saint Lucia and abroad. Fisher exact tests were conducted to assess association between qualitative variables. An a priori alpha level of 0.05 was used to determine statistical significance. Statistical analysis was performed with SAS 9.4 software (SAS Institute, Cary, NC, USA).

## 3. Results

### 3.1. Challenges

Given limited funding, staffing with investigator(s) with assists from university students was difficult. Steering committee members and field personnel had primary responsibilities competing with their study duties, and student volunteers were present only intermittently. As a result, we had periods of inactivity. Interviews were suspended mid-March to June 2020 due to the Covid-19 pandemic. This even meant some completed questionnaires remained inaccessible at the study office for months, delaying data entry. Further, due to limited human resources, collecting anonymized data from medical records was not possible, despite authorization from participants to obtain such data. 

### 3.2. Feasibility

Figure 1 is a summary flow chart for participant inclusion. Of the 67 eligible cancer survivors, 12 were from FOCS, 15 were from health care establishments, and 39 were from key informants. Four subjects refused, and 13 agreed to participate but were not interviewed. The final analysis was on 50 survivors (key informants = 39, FOCS = 10, and hospitals/clinics = 1). This was a response rate of 75%.

A little over half of the cancer survivors were women with breast cancer (*n* = 26) (Table 1). Seventy percent resided in urban areas (Castries and Gros-Islet), with others residing in smaller towns (Soufriere, Micoud, and Vieux-fort).

Table 2 displays information pertaining to interview quality. All interviews were conducted with the survivors themselves with the exception of four persons, among which two were deceased. Four participants were incident cases interviewed within the first 3 months of their diagnosis. Most participants interviewed had already completed all their initial active treatment (67%), whereas a smaller proportion was either still on treatment (20%) or did not start at all (12%). 

### 3.3. Patient Characteristics

Table 3 provides a comparison of sociodemographic and clinical characteristics by data source. Sixty percent of participants reported early stage cancer (Stage I and II), while seven did not provide an answer or did not know their stage. Compared to key informants, FOCS participants by point estimates were slightly younger (*p* = 0.32), more frequently female (*p* = 0.25), and hypertensive (*p* = 0.06), none of these being statistically significant. However, they had experienced much longer survivorship (*p* = 0.0014), while being with greater secondary education (*p* = 0.02), and, perhaps surprisingly, with more advanced stages of cancers (*p* = 0.16, not statistically significant). 

### 3.4. Cancer Care Pathways

Figure 2 maps pathways in diagnosis and treatment, and the time durations between events or decisions. Out of 50 survivors, six discovered their cancers incidentally. Concerning others, following the appraisal of first symptoms, 66% immediately considered their symptoms in need of medical attention, and so at least had the intention to contact a HCP. 

In total, 183 consultations were recorded for these 50 patients. General practitioners were involved in 22% of all investigations, and only 52% of first presentations. Four consultations were emergency room visits, three (6%) being first presentations. Of the 159 tests and investigations reported by participants, 111 were done in Saint Lucia, 46 abroad, and 2 were with an unspecified location. Turnaround times for pathology reports were significantly greater in Saint Lucia than those completed abroad (*p* = 0.0013). Among countries visited overseas for tests, the most frequent were Martinique (39%) and the USA (23%). Approximately 50% of investigations were completed in under a month, but a quarter of investigations exceeded 2 months until resolution/reporting (overall median = 25 days, IQR = 4–64 days).

Fifty-six percent of participants traveled outside of Saint Lucia for care, the most frequent destinations being the USA (29%), Martinique (13%), and Guyana (13%). This outbound medical tourism was driven by both system-related factors and personal factors (Table 4). There were six participants who were forwarded to services for palliative care and symptom control. Approximately half of all participants initiated treatment more than one month after diagnosis (overall median = 32 days, IQR = 19–86 days). Appendix A show the details for this section.

### 3.5. Perception of Delays and Care Experience

Participants were asked open-ended questions regarding delays between major events and were solicited for suggestions to reduce delays. Few participants were concerned about delays in test turnaround times (*n* = 4) or the pre-treatment interval (*n* = 13). There were few expressed concerns to improve the way investigations (*n* = 13), tests (*n* = 3), and explanations (for the patient, *n* = 15) were handled by HCPs. Thirty-three participants (76%) rated their overall care experience as at least “good”, two (4%) as fair, and eight (20%) as “poor” or “very poor”.

### 3.6. Funding of Cancer Care

Forty participants provided information on personal and other sources of funding for care. The most frequent source was family and/or friends (60%), and was considerably higher compared to payment using personal funds/savings (38%) and assistance from the government (15%). Other sources were medical insurance (38%), loan/credit (30%), fundraisers (30%), help from employers (10%), and donations from NGOs (8%), with the church community least (3%).

## 4. Discussion

We describe, for the first time, cancer care delivery in Saint Lucia from a patient perspective. We have tabulated information from 183 HCP consultations, 159 diagnostic tests, and numerous qualitative responses on the experience from 50 cancer survivors. Study findings include significantly longer turnaround times for biopsy results in Saint Lucia compared with biopsies done overseas. This study reveals frequent travel for cancer treatment, major funding from family and friends, but high satisfaction in the overall care experience.

This study demonstrated some local (not strictly “national”) capacity to study cancer health services, but we encountered many challenges conducting this first-of-its-kind study in Saint Lucia. A lack of research infrastructure and staffing are typical of LMICs in health research [35,36]. Given a low resource setting, we had limited capacity to manage the logistics and monitoring of field operations as compared to more developed countries, that benefit substantially from greater research administration, institutional support, and data management systems (e.g., more integrated clinical databases) [35].

This study has a number of methodological limitations; results should be interpreted with caution. First, in terms of our sampling, there is likely a selective survival bias. The proportion of early stage cancers was higher than expected, and probably due to sampling some survivors far after initial diagnosis [32]. Despite difficulties to generalize findings, and inherent flaws due to convenience sampling [31], we did have a representation of diverse patient experiences, including recently diagnosed cases (<3 months), long-time survivors, and persons living in remote rural communities. This is a strength of this study. Additionally, our assembled sample is representative of the most common cancer sites by sex for Saint Lucia [1,10]. Our findings of medical travel to Martinique coincide with findings in a prior study [37]. Further, patients from all major hospitals were represented in our key informant group.

Another limitation is that we used self-reported data, similar to a study in Haiti [29]. Information bias is possible, particularly for cancer stage, or in describing initial symptom severities, as severe symptoms and low stage could be viewed as inconsistent reporting (severe symptoms at first presentation including lumps, pain, weight loss, etc.) [38,39,40]. Furthermore, we suspect erroneous responses for stage resulting from a lack of health literacy among some participants. This includes notable missing data for stage and underreporting of system flaws. Participants also gave many partial dates or omitted them during interviews. Fortunately, the number of days between events was sufficient to reconstitute dates in most cases. Date of diagnosis was determined based on participants’ interpretation, rather than guidelines-consistent recommendations for this to be the date at which a histological determination concluded there was invasive cancer [32]. Dates of diagnosis did correspond to the date participants were told they had cancer, in most cases, which may distort the estimates for time intervals. However, this distortion is likely minimal as diagnosis announcements typically occur quite soon after the biopsy. Nevertheless, our questionnaire data provided some advantages compared to database studies: treatment modalities and location were more likely to be accurate [41], and we collected variables on self-appraisal, help-seeking, follow-up care, and opinions. 

Last, terms and measures in this research field are complex. Our study instrument would ideally require theoretical validation and established studies of validity. We tested it on two cancer survivors before starting this pilot [32]. Definitions for key events were based on the model of pathways to treatment [33], strengthening the comparability with other studies. Further, we conducted face-to-face in-depth interviews, and this reduces issues related to the interpretation of different terms and time intervals commonly associated with self-completion questionnaires without interviewers [42]. 

Building robust cancer care systems in SIDS requires substantial investment and prioritization based on careful assessment of the implications for patients. It is apparent that Saint Lucia urgently needs national or several good, local (i.e., regional) clinical databases [9] to guide research and cancer plans, including a focus on data analytics to discover priority pathways for improvement, and dissemination and implementation science to identify patient- and public-centered solutions (e.g., in patient navigation, multi-disciplinary case conferencing, and so on). Networking with regional SIDS can help strategizing towards greater cancer advocacy, sustainable research, and cancer care more appropriate for SIDS (e.g., regionalization of some services, localization of others). Priority setting for change is far more important in resource-limited contexts, such as SIDS, so greater effort is needed to understand the context and sort the many possible options for development [9] into those that are of greatest importance. Importantly, data as obtained in this study can be combined with clinical databases to provide a comprehensive picture as a basis for system planning, action, and further research. Studies that include hundreds of patients may be more representative, and motivating for change, than smaller studies. 

## 5. Conclusions

This is the first community-based study on cancer health services in Saint Lucia from a patient perspective. We mapped main pathways to diagnosis and treatment, and estimated time intervals between key events. Access to cancer care is complex due to frequent medical traveling that introduces inequalities in management. Survivors accessed health services overseas (e.g., USA, Martinique, and Guyana) for motives other than just availability on-island. Funding for care was mostly secured by family and friends, with limited support from the government. We observed evidence of low health literacy and limited awareness of health care standards among survivors, which likely reduces their ability to use health services in an efficient, effective, and safe manner. To reduce system-related mortality in the region, patient-centered implementation research is needed, along with inter-state comparisons, and regional studies. 

An in-depth analysis of interviews using mixed-methods approaches is planned. We also plan to collect data from medical records and compare with our self-reported data to assess patients’ ability to recall accurately their care. Our studies will provide a foundation for future related studies by generating new data for sample size calculations as well as uncovering new research questions. 

## Figures and Tables

**Figure 1 ijerph-18-04770-f001:**
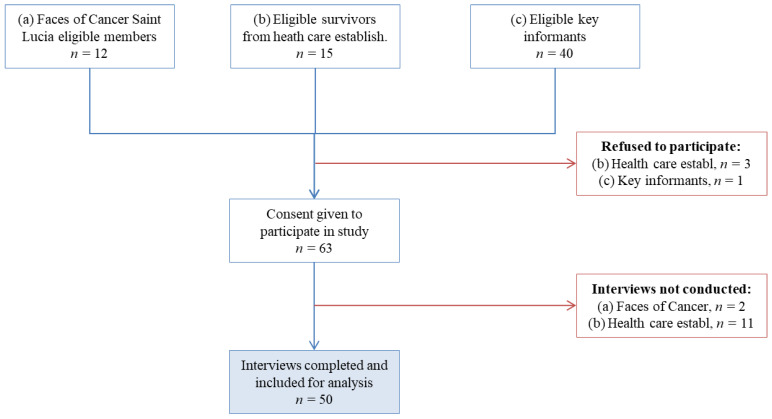
Flow chart of subject inclusion for pilot survey on cancer pathways.

**Figure 2 ijerph-18-04770-f002:**
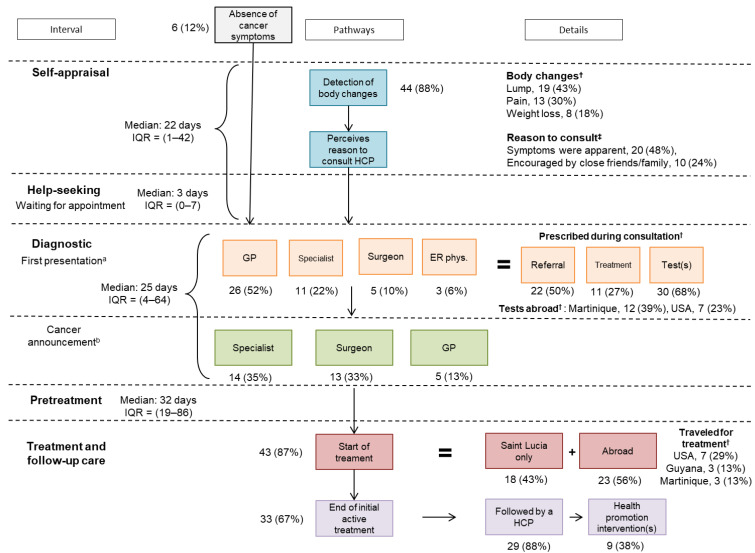
Summary of pathways to diagnosis and treatment of Saint Lucian cancer survivors with time intervals. Saint Lucia (West Indies), 2019–2020. HCP: health care provider, GP: general practitioner, ER phys: emergency room physician, Specialist: gynecologist, urologist, and specialist (unspecified), IQR: Interquartile range. ^†^: Possibility of having several responses; hence, sum of values may not correspond to the number of patients included; see Appendix A for details. ^‡^: other reasons account for 28%, see Appendix A for details. a: Symptomatic and incidental cases included (*n* = 50). Other health care providers at first presentation account for 13%, including: nurse, primary care (unspecified), radiologist, and oncologist. b: Other health care providers who gave diagnosis announcement account for 19%, including: oncologist, radiologist, emergency room physician, doctor’s assistant, and unspecified.

**Table 1 ijerph-18-04770-t001:** Sociodemographic characteristics of cancer survivors by cancer site.

Characteristics	Overall		Breast		Female Pelvis ^a^		Prostate		Other ^b^		*p*
	*n* = 50	%	*n* = 26	%	*n* = 10	%	*n* = 9	%	*n* = 5	%	
**Sex**											<0.0001
Male	13	26	0	0	0	0	9	100	4	80	
Female	37	74	26	100	10	100	0	0	1	20	
**Age at diagnosis**											0.05
<50	15	30	10	38.5	3	30	0	0	2	40	
50–65	26	52	14	53.8	6	60	4	44.4	2	40	
>65	9	18	2	7.7	1	10	5	55.6	1	20	
**Survivorship**											0.39
0–4 months	4	8.2	1	4	3	30	0	0	0	0	
5 months–1 y	11	22.4	5	20	2	20	2	22.2	2	40	
2–3 y	11	22.4	6	24	1	10	4	44.4	0	0	
4–5 y	7	14.3	2	8	2	20	2	22.2	1	20	
6–9 y	11	22.4	7	28	2	20	1	11.1	1	20	
10 + y	5	10.2	4	16	0	0	0	0	1	20	
Missing	1		1		0		0		0		
**Stage at diagnosis**											1
Early (I/II)	26	60.5	16	61.5	5	55.6	4	66.7	1	50	
Advanced (III/IV)	17	39.5	10	38.5	4	44.4	2	33.3	1	50	
Missing	7		0		1		3		3		

Saint Lucia (West Indies), 2019–2020; ^a^ cervix *n* = 3, endometrium *n* = 5, ovary *n* = 2; ^b^ colon (3 men), parotid gland (1 woman), and leukemia (1 man).

**Table 2 ijerph-18-04770-t002:** Description of interview quality and acceptability.

Category	*n*	%
**Interviewee**		
Patient	46	92
Next-of-kin	4	8
**Vital status**		
Alive	48	96
Deceased	2	4
**Time between diagnosis and interview**		
Median (y), IQR	3.4	1.5–7
Treatment status		
Finished initial active treatment	33	67.4
Still on treatment	10	20.4
No treatment taken	6	12.2
Missing	1	
**Interview length**		
Mean (SD). Min Max	1:24 (0:35)	0:37–2:49
**Interview quality rating ^†^**		
Poor/Mediocre	0	0
Good	22	48.9
Very Good	15	33.3
Excellent	8	17.8
Missing	5	

Saint Lucia (West Indies), 2019–2020; ^†^ interview rated by field investigator.

**Table 3 ijerph-18-04770-t003:** Sociodemographic characteristics of cancer survivors and comparison between data sources.

Characteristic	Faces of Cancer		Key Informants ^†^		*p*
	*n*	%	*n*	%	
**Sex**					0.17
Male	1	10	12	30	
Female	9	90	28	70	
**Cancer stage**					0.16
I	2	20	10	30.3	
II	2	20	12	36.4	
III	6	60	7	21.2	
IV	0	0	4	12.1	
Missing	0		7		
**Age at diagnosis**					0.32
<50	4	40	11	27.5	
50–65	6	60	20	50	
>65	0	0	9	22.5	
**Survivorship (y)**					0
<7	2	22.2	33	82.5	
7–10	5	56	4	10	
>10	2	22.2	3	7.5	
Missing	1		0		
**Marital status**					1
Single	5	50	18	46.2	
Married	3	30	13	33.3	
Divorced/Separated	1	10	3	7.7	
Widowed	1	10	5	12.8	
Missing	0		1		
**Education level**					0.02
Primary	1	10	15	38.5	
Secondary	7	70	8	20.5	
Tertiary	2	20	16	41	
Missing	0		1		
**Private medical insurance**					0.46
Yes	5	50	13	32.5	
No	5	50	27	67.5	
**Hot water at home**					0.50
Yes	6	60	18	46.2	
No	4	40	21	53.9	
Missing	0		1		
**History of medical condition(s)**					0.29
Yes	7	70	18	45	
No	3	30	22	55	
**Professional status**					0.52
Still working	4	44.4	20	51.3	
Retirement/Volunteer	5	55.6	12	30.8	
Unemployed	0	0	4	10.2	
Invalidity due to sickness	0	0	3	7.7	
Missing	1		1		

Saint Lucia (West Indies), 2019–2020.; ^†^ survivor from hospital setting mixed with the key informant group.

**Table 4 ijerph-18-04770-t004:** Frequency of motive for seeking care outside of Saint Lucia.

Motive for Choice of Country of Care	Diagnostic Test		Treatment	
	*n* = 26	%	*n* = 22	%
Attracted by the price	6	23.1	8	36.7
Personal preference for location	10	38.5	13	59.1
Recommended by someone ^a^	5	19.2	6	27.3
Referral ^b^ by HCP	9	34.6	8	36.4
Location of specific lab/hospital	1	4	0	0
Proximity to family/close friend(s)	9	34.6	9	40.9
The service accessed was not available in Saint Lucia *	8	30.8	8	36.4

Saint Lucia (West Indies), 2019–2020; ^a^ Someone else excluding patient’s HCP (e.g., family/friends); ^b^ Formal referral from patients’ HCP; * Treatments reported as unavailable in Saint Lucia brachytherapy and radiotherapy. Diagnostic tests reported as unavailable: MRI, PET scan, CT scan, surgery.

## Data Availability

The datasets analyzed during the current study are available from the corresponding author upon reasonable request.

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
