# Peer review of "Difficulties in Accessing Cancer Care in a Small Island State: A Community-Based Pilot Study of Cancer Survivors in Saint Lucia"

_ijerph, 2021, doi:10.3390/ijerph18094770_

Round 1
Reviewer 1 Report
This is a well written and comprehensive description of the first evaluation of cancer survivorship in St. Lucia. The authors present new data among a small sample of cancer survivors and also highlight implementation factors relevant to undertaking research in a Small Island Developing State. The goals seem to be to two-fold: presenting patient-level cancer data and helping to provide context through which decisions about increasing comprehensive cancer care can be achieved in the community. Some specific questions and comments follow:
Abstract: some additional clarity would be useful in the Abstract regarding the data collection methods. As is, it was not obvious how this was a retrospective cohort study, it sounded like a cross-sectional study of cancer survivors.
Introduction:
- Please provide a definition of “out-bound medical tourism” for those who are unfamiliar with this terminology.
- It would be useful to present cancer mortality information (and cancer incidence) in the form of rates
Methods:
- The phrase “P-values below 5% were considered statistically significant” would be more appropriately worded as something like, “An a-priori alpha level of 0.05 was used to determine statistical significance”
Results:
- Figure 1. The two excluded boxes are confusing. Specifically, it isn't clear what the two rows of exclusions mean in each box.
- Table 1. Would be nice to have an overall column as well in this table instead of Table 3 to provide the description of how this population generalizes to the distribution of cancer survivors in St. Lucia.
- Table 2. How was interview quality rated?
- Figure 2. This was a nice but also difficult figure to follow. In particular, it is hard to follow each denominator and to understand the flow of any missing data. For example, for the row "diagnostic" the sum of the percent who saw "gp", 'Specialist", "Surgeon" and "er phys." does not add to 100%. The sum for these bins is 45, which confused me as it seemed like these participants would be flowing from the 44 who reported a detection of body change. It might be useful to provide a companion table as well if confusion persists.
- Table s10 might be worth including in the manuscript and elaborating on as understanding the reasons for going aboard for care might help inform "gaps" to providing comprehensive cancer care locally.
Discussion
- Unfortunately, there was not opportunity in this study for the medical record review. Given that there was some missing data and concern about health literacy in the study population, another next step not articulated in the manuscript could be doing the medical record review. This would allow a comparison of results using that standard to the imputed information used in this study and could also be useful by describing how well cancer survivors with lower health literacy are able to recall and describe their care.
Reviewer 2 Report
This is well-written paper with detailed, logic and sound research methods. They have demonstrated the potential challenge and feasibility to conduct research among cancer survivors and difficulties in accessing cancer care in a small island.
The only comments I have for this study is the sample size is relatively small. Given the difficulties in studying and recruiting patients from this island, it is understandable that enrolling a large number of participants are unlikely within a short period of time.
